# With or without a Vaccine—A Review of Complementary and Alternative Approaches to Managing African Swine Fever in Resource-Constrained Smallholder Settings

**DOI:** 10.3390/vaccines9020116

**Published:** 2021-02-02

**Authors:** Mary-Louise Penrith, Armanda Bastos, Erika Chenais

**Affiliations:** 1Department of Veterinary Tropical Diseases, Faculty of Veterinary Science, University of Pretoria, Onderstepoort, Pretoria 0110, South Africa; 2Department of Zoology and Entomology, Faculty of Natural and Agricultural Sciences, University of Pretoria, Pretoria 0028, South Africa; adbastos@zoology.up.ac.za; 3Department of Disease Control and Epidemiology, National Veterinary Institute, S-751 89 Uppsala, Sweden; erika.chenais@sva.se

**Keywords:** biosecurity, disease control, disease prevention, genetic resistance, livelihoods, subsistence pig farming

## Abstract

The spectacular recent spread of African swine fever (ASF) in Eastern Europe and Asia has been strongly associated, as it is in the endemic areas in Africa, with free-ranging pig populations and low-biosecurity backyard pig farming. Managing the disease in wild boar populations and in circumstances where the disease in domestic pigs is largely driven by poverty is particularly challenging and may remain so even in the presence of effective vaccines. The only option currently available to prevent ASF is strict biosecurity. Among small-scale pig farmers biosecurity measures are often considered unaffordable or impossible to implement. However, as outbreaks of ASF are also unaffordable, the adoption of basic biosecurity measures is imperative to achieve control and prevent losses. Biosecurity measures can be adapted to fit smallholder contexts, culture and costs. A longer-term approach that could prove valuable particularly for free-ranging pig populations would be exploitation of innate resistance to the virus, which is fully effective in wild African suids and has been observed in some domestic pig populations in areas of prolonged endemicity. We explore available options for preventing ASF in terms of feasibility, practicality and affordability among domestic pig populations that are at greatest risk of exposure to ASF.

## 1. Introduction

African swine fever (ASF) is a lethal haemorrhagic fever of pigs and Eurasian wild boars (*Sus scrofa*) that evolved in south-eastern Africa in a sylvatic cycle between common warthogs (*Phacochoerus africanus*) and argasid ticks of the *Ornithodoros moubata* species complex that live in their burrows [1,2]. Movement of pigs and pork has resulted in the disease becoming widespread in sub-Saharan Africa wherever pigs are produced, as well as introductions to other continents [3,4]. The introduction of an East African, genotype II ASF virus into the Caucasus in 2007 [5] resulted in the disease becoming widespread in eastern and central Europe and, since 2018, in China and many other countries in Asia [5,6,7,8,9,10,11,12,13]. This current genotype II epidemic involves both domestic pigs and wild boars, although in most parts of Europe the infection is maintained in wild boar populations without involvement of domestic pigs [14].

An effective ASF vaccine has been the holy grail of ASF research and control for more than a century. It would be an invaluable tool to manage ASF and perhaps to eradicate it from non-endemic areas [3,15,16]. In spite of recent considerable progress towards a safe and efficacious vaccine, until such vaccines can be made widely available alternative control measures remain essential to enable sustainable and profitable pig production. A combination of vaccination and enhanced biosecurity would probably offer the best protection [16,17]. In the areas of southern, central and eastern Africa where ASF is endemic, it is likely that alternatives to vaccination may always offer the best option for sustainable disease control. This is because the majority of the domestic pigs in these areas are raised in free-range systems by poor farmers on small-scale holdings, making successful vaccination a challenge. Furthermore, the considerable genetic and serotypic diversity of the virus in this region and the presence of a sylvatic cycle pose additional challenges for vaccine development and virus eradication, respectively [18,19,20]. Elimination of the virus is not an option, as the sylvatic cycle involves African wild suids that are impervious to the pathogenic effects of the disease, and a highly cryptic and long-lived arthropod vector. Apart from the fact that eradication of wildlife is ethically and ecologically unacceptable, most countries in the endemic region derive high revenues from nature-based tourism and there is an increasing emphasis on wildlife protection [21,22,23].

Eradication of ASF in domestic pigs, even without a vaccine, is feasible, having been achieved for the genotype I virus introduced into the Iberian Peninsula in 1960 and is on its way to being achieved in some of the EU countries affected by the current genotype II outbreak strain that was introduced into Georgia in 2007 [5]. In Belgium, Czech Republic, Hungary and Germany only wild boars have been infected and in both Czech Republic and Belgium the infection has been eradicated [24,25,26]. Recent studies have demonstrated an encouraging decline in ASF virus circulation in wild boars in Estonia and Latvia as well [27,28,29]. On the other hand, in some eastern European countries, including Bulgaria and Romania, the situation is less favourable. This is also the case in sub-Saharan African regions where ASF has become endemically established in domestic pigs despite the absence of a sylvatic cycle [30]. The common denominator appears to be the preponderance of mainly small-scale, backyard and village pig production. In Sardinia the presence of ASF genotype I for more than 40 years can be ascribed to circulation in poorly managed free-ranging pigs [31]. Reports to the World Organisation for Animal Health (OIE) indicate that the great majority of pig farms reporting ASF in the Asia-Pacific region are backyard or village operations. This region is also a candidate for alternative approaches to vaccination that should be initiated as soon as possible. As in Africa, most of these countries’ resources are limited both at national level and amongst the smallholder farmers, so vaccination even when available may not be the most sustainable option. Most alternatives to vaccination will involve pig management practices and biosecurity.

Although not a short-term solution, exploitation of innate resistance to the pathogenic effects of the virus may offer a sustainable solution for managing ASF, particularly where selection for high productivity traits is not important, as in free-ranging domestic pigs and potentially also in wild boars. Infection of African wild suids with ASF produces no clinical signs of disease [32,33,34], and although rare, individual domestic pigs and wild boars [35] also exhibit resistance to the pathogenic effects of the virus. For the purposes of this review, we define an ASF-resistant animal as one that following infection is able to limit pathogen burden such that no discernible negative health impacts are observed, whereas an ASF-tolerant animal is one that, following infection, displays negative health impacts due to continued activity of the pathogen, but does not succumb to the disease. The proportion of pigs that do not develop clinical signs but seroconvert is much higher in certain populations in endemically infected areas in Africa [36,37]. Limited studies on such pig populations have indicated that they can be infected by more than one genotype of highly virulent viruses without developing any clinical signs and that infectivity is of relatively short duration [37]. They are therefore unlikely to become long-term carriers, i.e., pigs that show no signs of infection but are able to transmit the virus through contact to other pigs [37,38].

This review examines potential alternatives to vaccination for achieving ASF control in resource-constrained, smallholder pig farming settings, a stratum of the pig sector that is at particularly high risk of ASF due to certain characteristics of the low-input husbandry systems. Considering that this pig production sector constitutes a high proportion of global pig production activities, with more than 80% of the pigs in many developing countries raised in the smallholder sector, approaches to improve implementation of feasible and sustainable biosecurity measures in traditional free-range and backyard pig production and in the value chains linked to these systems is crucial. In contrast to vaccination, these measures can be implemented immediately, to great effect. Most of the examples given and references used are from Africa, where ASF originated and has been known as an endemic disease for a century, and where large amounts of the research cited in this study have been generated. However, the measures proposed are not area-specific and could be applied widely in small-scale pig production. In particular, backyard and village farmers have featured prominently in reports of ASF outbreaks in the recently-affected Asia-Pacific region. Studies on ASF epidemiology and control in the smallholder sector from this region are just starting to emerge, with evident parallels to the same farming sectors in Africa. We also consider innate resistance to ASF as a complementary approach to improved pig production in the smallholder section in the longer term.

## 2. Pig Husbandry Systems at High Risk for African Swine Fever

In the presence of ASF, any pig population of known susceptibility (i.e., *Sus scrofa*) can be considered to be at risk. Until effective vaccines become available, alternative disease control measures are needed in all types of susceptible pig populations, from pigs in sophisticated industrial domestic pig production systems to free-ranging pig populations that include domestic pigs, feral pigs and Eurasian wild boars. The risk to domestic pigs that are permanently confined varies according to the level of management, while the risk to free-ranging populations always will be higher. Modern industrial commercial pig farms and feral pig/wild boar populations are not covered in the present review. Commercial outdoor pig production is mentioned but the focus is on small-scale, non-industrial pig production systems. Since smallholder pig production is frequently associated with informal marketing systems that may pose a high risk for transmission and spread of ASF beyond the initial outbreak focus, the potential for identifying and mitigating ASF along value chains is briefly considered [39].

### 2.1. Production Systems with Free-Ranging Pigs

Outdoor husbandry varies from traditional free-ranging pig production systems at village level, especially in Africa and the Asia-Pacific region, to some modern types of pig production in wealthy countries, including organic outdoor production and the production of high-quality products such as Iberian hams [40,41]. Biosecurity for commercial outdoor/organic pig production, a growing modern trend resulting largely from consumer concerns for animal welfare, differs from that appropriate for indoor farming [40]. The pigs enjoy varying degrees of freedom of movement, but the premises are usually protected by a pig-proof perimeter fence and other biosecurity measures are in place concerning the hygiene of the environment [42]. Biosecurity in the traditional “dehesa” or ”montados” outdoor pig production systems in the south-western Iberian Peninsula where high quality cured pork is produced, was improved after losses were experienced during the ASF presence in the Iberian Peninsula from 1960 to the early to mid-1990s [43,44]. The main focus of the biosecurity measures in the “dehesa” production system is on a strong perimeter fence, which is sound in principle but seldom applicable in resource-poor settings. This review, while supporting the principle of fencing and recommending it in situations where it is practical and feasible, examines other measures that can be helpful when permanent confinement of pigs is not an option.

In rural areas in many African countries where ASF is endemic, the majority of pigs are kept by poor people owning low numbers of pigs (generally <15 animals), trading in local markets and practising fully or partially free-range systems with varying degrees of management inputs [45,46,47,48,49,50,51,52,53,54]. In spite of their disadvantages, free-ranging systems are necessary for the livelihood of millions of poor smallholders [55,56,57]. Pigs provide an easily available cash income with rapid generation rate, constitute an important and affordable protein source and might in addition have other purposes such as provision of fat for making soap, manure as a fertilizer, serving as a living bank account and sometimes being used for ceremonial purposes [51,52,54,58]. Similar situations exist in the Asia-Pacific region [59,60,61,62]. Research has demonstrated that pigs are kept in free-range systems for economic reasons even though the owners understand the disadvantages of the system including the health risks for pigs and people [55,63,64,65]. Challenges for vaccination in this stratum of the pig population wherever it occurs can include lack of financial resources on the part of pig producers and governments, difficulties of distribution and implementation in remote areas with poor infrastructure posing challenges for access and cold chain observance, logistic challenges in supplying low numbers of doses to scattered small herds of pigs, and the high turn-over of pigs posing a need for frequent re-vaccinations [3,65]. In some communities the relatively low economic value of pigs contributes to reluctance to invest in pig health management [66]. Similar challenges have been discussed for the recently launched global eradication campaign for pest de petits ruminants [67,68]. In addition, in countries with the classic sylvatic cycle with multiple genotypes, vaccines that achieve broad coverage would be necessary [69] and these may never be prioritized or developed due to the relatively small market.

### 2.2. Backyard Pig Farms

Small scale and backyard pig farms have been prominently involved in the outbreaks of ASF in Eastern Europe and Asia [6,70,71,72,73] as well as in Africa, where large-scale commercial pig farms are relatively rare [30,74]. Periodic release of confined pigs to scavenge may contribute to the involvement of backyard farms [70], in which case the challenges are those posed by free-ranging pigs. When the pigs are permanently confined and must be fed, food waste fed as swill is often the most important source of infection [8,75], particularly in urban and peri-urban conditions where leftover food from commercial food outlets is easily available [76]. Food waste containing infected pork fed to pigs was considered to be the most like source of the initial outbreaks in China [13] and Mongolia [8]. However, in an outbreak in a backyard pig farm in Bulgaria in pigs that were permanently securely confined and fed commercial feed, the source of infection was believed to have been fomites introduced via people who had unrestricted access to the premises, including to care for the pigs in the absence of the owners [77]. Sale of pigs through agents who move from farm to farm collecting pigs has also been identified as a risk factor for introduction of diseases like ASF [78]. A recent study in Tanzania in areas where smallholder pig farming predominates [48] also identified unrestricted access to pigs as well as fomites as important risks for ASF transmission [79]. In addition to some of the above risk factors, visits from professionals, proximity to outbreaks in wild boars and feeding contaminated forage to domestic pigs were associated with outbreaks of ASF in backyard pig farms in Romania [71]. In Latvia, breaches in biosecurity and feeding contaminated crops or grass in areas where infected wild boars had died were identified as causes of outbreaks in backyard farms [73].

## 3. Biosecurity

Awaiting the holy grail of an ASF vaccine, improved biosecurity is still the only way to achieve ASF prevention, stop transmission and control outbreaks. Biosecurity has been defined as “a set of preventive measures designed to reduce the risk of transmission of infectious diseases in crops and livestock, quarantined pests, invasive alien species, and living modified organisms” [80]. In the context of this review we refer to biosecurity on farm level, i.e., measures aiming to prevent ASF from entering into a farm or a population, and to reduce transmission between individuals or groups of individuals once introduced. To design such measures, or advise on their use, certain properties of the biological agent in question need to be known. Since the first description of ASF [81] almost a century of research has focused on its virology, pathology and epidemiology [16], and enough of the biological aspects of ASF to design technically effective biosecurity measures have been known for a long time.

Notably, not all of the known sources of virus and routes of transmission are important in all contexts (Table 1). High loads of ASF virus in blood and excretion of infective quantities of ASF virus by acutely infected domestic pigs and wild boars provide the most potent source of infection for conspecifics [82,83].

The virus can easily be neutralized by heat-treatment, or, after removal of organic material masses, by a number of standard disinfectants [81,84,85]. The only published information on durability of the virus in meat relates to various processed products [86,87,88,89] but it can be expected to remain viable for weeks or months in fresh chilled or frozen pork. The transmissibility, often measured as R_0_, of ASF is relatively low, especially between farms or between groups of feral pigs or wild boars [90,91,92]. If pigs get into direct oro-nasal contact with infected blood, however, R_0_ can be high [92]. These viral and epidemiological features thus mean that ASF transmission can be reduced by improving biosecurity on and between farms, in critical activities such as trade and slaughter, and in application of heating and disinfection to inactivate the virus. The biosecurity measures can further be rather focused, aiming to reduce direct contact between domestic pigs of unknown disease status and to minimise contact between pigs and infected blood or pig products [3,16,93].

The importance of human actions (legal and illegal) in long-distance transmission of the virus as well as farm-level transmission is well recognized [7,77,79,92]. It thus seems that more, or other, aspects than the biological properties of the virus need to be addressed in order to improve implementation of biosecurity.

### 3.1. Biosecurity Recommendations for Different Levels of Pig Production Systems

The biosecurity systems that are usually implemented on large-scale modern commercial pig farms are more than sufficient to protect against ASF provided a culture of adherence to the measures prevails [94]. In areas where ASF is endemic, compartmentalization of commercial pig farms is highly recommended [95], as it usually enables trade to continue regardless of the status of the country or area. One of the main reasons for undertaking this study was to provide alternatives to the ‘one size fits all’ approach sometimes recommended in developed countries. The biosecurity measures recommended for modern commercial farms are not feasible in resource-poor smallholder settings and, based on our knowledge of ASF, prevention can be achieved with a simple set of measures that can be agreed with the pig farmers.

The highest risk of ASF is posed by direct contact with infected pigs or, where present, wild boars, contact with their secretions and excretions via fomites, and ingestion of meat or other tissues including blood from infected pigs (Table 1). Preventing these contacts needs to be the focus while designing risk-based biosecurity measures for smallholder pig producers that are feasible, affordable, socio-culturally acceptable and cost-effective.

Ticks of the genus *Ornithodoros* are competent biological vectors of ASF. The risk of outbreaks being initiated by the bite of an infected warthog-associated tick is limited to areas with an interface between warthogs, ticks, and domestic pigs. Outbreaks in domestic pigs that are allowed to roam freely in such areas are frequent [102]. *Ornithodoros* ticks infesting pigsties are able to maintain the infection for long periods, even in the absence of infected pigs, and have been incriminated in initiating outbreaks [103,104,105]. However, in the great majority of areas now affected by ASF they are not known to be a concern. The remaining potential transmission routes that have been investigated appear to pose a low risk and investing in preventing them would likely not be cost-effective. Airborne transmission has only been demonstrated to occur over short distances in closed housing [106,107]. To date, the only blood-feeding arthropod demonstrated to be a competent mechanical vector of ASF virus is the stable fly, *Stomoxys calcitrans*, which was able to infect pigs during a blood meal one and 24 h after ingesting infected blood [99]. Another study showed transmission of the virus to pigs fed on flies for up to 12 h after they had ingested infected blood [100]. However, its actual importance in the epidemiology of ASF is likely to be minor due to the short time window during which infectious virus is present and limited dispersal potential. Long distance dispersal by wind has been reported, but with much shorter distances recorded for ”partially fed and unfed” flies [108]. Investigation of other blood-sucking arthropods as well as blowfly larvae has failed to provide any evidence of potential involvement in transmission of ASF virus [96,101,109,110].

The biosecurity measures that prevent the common transmission routes are straightforward and should be feasible for pigs that are not allowed to roam freely (Table 2). However, to be successfully implemented, biosecurity recommendations must be based on a thorough understanding of the local context in which they should be applied, including aspects such as feasibility and acceptability based on culture, tradition and situated knowledge [111,112,113]. To meet these criteria, promoting participation of end-users in the design and adaptation of the measures they are meant to implement is highly recommended [63,111,112]. The identified socio-economic and cultural factors that can hamper implementation of biosecurity are briefly enumerated (Table 3) and possible solutions for the most basic measures, namely confinement and provision of safe feed, are discussed in more detail.

### 3.2. Confinement of Pigs

For confined pigs to thrive, the shelter should provide adequate shade and protection from exposure to adverse weather conditions, and the owner should be able to provide sufficient feed, water and hygiene [114]. Poor farmers practising subsistence farming with free roaming pig management systems in which pigs can be raised with little or no investment or expenditure sometimes view providing pig-proof housing, feed and water as a set of new and insuperable challenges. If demand and prices in local markets are unlikely to rise in response to investment in better pig husbandry, i.e., farming is not driven by market forces, investing in pigs may not be worthwhile [55,56,57,63,115,116]. Nevertheless, confining pigs can be encouraged through participatory identification of affordable local construction materials for housing and local sources of feed, combined with discussions concerning advantages of pig confinement for protecting humans against porcine cysticercosis and pigs against ASF, predation, theft, retaliation for damaged crops and road accidents. Local culture and traditions can also hinder application of biosecurity. Denying neighbours free access to the yard or asking them to wash hands before entering might for example be absolutely unacceptable in some cultures or communities [117]; (EC, personal observations). In-depth discussions with communities about preventing ASF at pilot sites in Timor-Leste after the incursion of ASF into the country allayed concerns about exclusion from properties with pigs being a hostile and unneighbourly action and enabled improved protection of pigs and reduced mortality due to ASF at the pilot sites [117].

Legislation relating to subsistence farming is rarely useful but can be successful under particular conditions. One example is in Mauritius, a tiny island where free-range pig keeping is prohibited due to an optimal land use policy. Certain private farms and community sites are designated for pig farming and regulations can be enforced due to the small geographical area involved, although some free-ranging pigs were reported during the 2007 ASF introduction [118]. Some examples were also observed at village level in Togo, Mozambique and Uganda [MLP and EC personal observations], where the head of the village or the villagers collectively decided that all pigs should be confined, and the villages were able to avoid ASF.

### 3.3. Safe Feed

Feed costs are the main determinant of profit at all levels of pig farming. Wherever pigs are produced at subsistence level, feed costs have to be kept as low as possible to enable a small profit. This imperative may result in pigs being left to fend for themselves or being fed on low quality feeds, a small proportion of which may pose a risk of diseases like ASF. Although the oral dose required to infect pigs with ASF virus is higher than the parenteral or intranasal doses, feed has proven to be a potent source of infection for pigs [81]. Infection can be achieved via the oral route with a lower dose in a liquid medium [81,119]. Various potential sources of infection by ingestion have been identified. Scavenging pigs have access to infected material in the form of carcasses of pigs or wild boars that have died of ASF or carelessly disposed waste that includes remains of infected pigs. Pigs may also be inadvertently infected by consuming vegetation contaminated with infected blood, saliva, urine or faeces [120]. However, ASF outbreaks are often traceable to infected material fed to the pigs by their owners [73].

Leftover food either from the table or obtained from outlets such as restaurants, hotels, hostels or hospitals has been an affordable feed solution for smallholder pig farmers worldwide. Particularly in peri-urban settings, catering waste that may contain remains of raw or inadequately cooked meat may be freely available and offer the simplest solution to feeding the pigs. In the absence of ASF, domestic and other locally obtained leftovers should theoretically be safe, but unfortunately a number of introductions of ASF (and other transboundary diseases) have been attributable to swill containing infected meat that originated elsewhere. Two of the most significant recent transboundary introductions of ASF virus have been reliably traced to ships’ galley waste disposed of on open landfills accessible to free-ranging pigs [5,121]. The introduction of ASF into the island of Mauritius in 2007 was believed to have resulted from galley waste accessed by pig farmers and fed to their pigs as swill, similar to the introduction of ASF into Brazil and several other countries during the last century [118,122,123]. Swill not linked to port or airport waste but containing legally or illegally imported infected pork was likely responsible for introductions into China and Mongolia [8,13].

Many countries have banned swill feeding, but compliance can only be guaranteed in commercial farms with production targets that necessitate feeding a high-quality balanced ration. An alternative to a total ban in some countries is legislation requiring boiling the swill for 30 min or even an hour, with stirring, a requirement that is excessive in terms of destroying the virus and is hardly more likely to be respected than a total ban. In developing and other countries where subsistence agriculture still exists or is prevalent, as well as countries where industrial pig farming is difficult in terms of space and/or feed requirements, it is increasingly recognised that banning swill feeding is impractical, and constructive approaches to ensuring its safety are required [124,125,126,127,128,129,130]. The approaches, which involve heating, dehydration and fermentation or combinations of two of those described in literature cited above, refer to the conversion of waste food to animal feed at a commercial level. However, there is no reason why turning swill into a safe and more nutritious feed for pigs should not be done at home or cottage industry level, including dehydration of the swill by sun-drying.

Feeding of contaminated vegetation to pigs can be avoided by visually identifying gross contamination with blood or excrement and rinsing with clean water to provide additional assurance. Heavily contaminated vegetation is most likely in the vicinity of an animal that has died of ASF, so vegetation from such areas should not be used.

For scavenging pigs, the minimum that can be done to prevent access to ASF-infected carcasses is to safely dispose of them. Deep burial or burning are the conventional recommendations for disposal of infected carcasses [131], with the latter often considered to be too expensive. In some communities, burial may be rejected due to lack of land or labour required [112] or because it is reserved for humans and may be prohibited as inappropriate for animal carcasses (EC, personal observations). In poor communities where meat is a luxury and pigs are kept for sale rather than home consumption, pigs that die are often butchered and eaten. It may even be strongly prohibited or a taboo to throw away meat [112], (EC, personal observations). Provided that the meat is well cooked and uncooked remains are not fed to the pigs, this can be considered safe disposal. In modern western culture cooking and consumption of infected animals is not considered acceptable [83] but whether the opinions of those who have never felt hunger or deprivation can or should be imposed upon those who have is questionable. However, the option of cooking and consumption should be limited to the owners of the pig, who are aware that the pig probably died of ASF, and those with whom they choose to share the well-cooked meat.

Wild boar carcasses have been identified as a considerable threat for sustaining circulation of ASF, where infection in wild boars is an issue measures are therefore usually in place to deal with the carcasses [6,14]. We know of no documented cases of environmental contamination by the carcasses of African wild suids, which usually do not contain significant amounts of ASF virus [21,34].

### 3.4. Trade and Slaughter

In the subsistence farming systems in ASF endemic countries in Africa trade and slaughter have been identified as important for disease transmission [75,76,115,132]. In case of pig disease or death, pigs may be traded live or slaughtered or butchered and sold to minimise the loss and potential shock to household economies [115,133]. Specifically, emergency sale of pigs by smallholder farmers has been identified as an important contributor to spread of ASF once outbreaks have occurred in an area [72]. Improving farmers’ ability to recognize the signs of does not necessarily prevent sale of infected pigs.

In the smallholder pig production value chains pig slaughter is mostly done at village slaughter slabs without veterinary inspection or the possibility of maintaining even basic slaughter hygiene. Such practices combined with frequent under-reporting of diseases [134] and high viral loads in blood make biosecurity improvements in trade and slaughter especially important for breaking the transmission chain [135]. Measures for this part of the value chain should aim at reducing the opportunities for pigs to come into direct or indirect contact with blood and offal. Only trading in healthy pigs and only slaughtering and butchering pigs that are alive and healthy are obvious recommendations. Slaughter should be performed on a surface that can be cleaned and in a way that does not disperse blood elsewhere, with safe disposal of inedible offal and blood, using protective clothing and boots while trading or slaughtering and separating the utensils used for slaughter from other utensils that might get into indirect contact with pigs.

## 4. Genetic Resistance

### 4.1. Wild African Suids

The ability to survive infection with virulent ASF virus without developing marked clinical signs was observed in warthogs and bushpigs (*Potamochoerus larvatus*) [32,34,81]. Experimental infection of both warthogs and bushpigs indicated that in both species virus titres in blood and tissues declined fairly rapidly in the first two months after infection and the virus was eventually eliminated [32]. Transmission to in-contact domestic pigs from warthogs did not occur [32,81,136] and transmission from bushpigs to domestic pigs occurred only in the period immediately post-infection and then only with one of two viruses used for the experimental infection studies [32]. These results indicate that the failure of the wild pigs to show adverse effects is the result of genetic resistance rather than tolerance to the virus and fuelled speculation that similar resistance might be possible in domestic pigs. Such resistant pigs would be an asset for pig breeders in ASF endemic areas and could contribute to the options available for minimising the impact of ASF [137].

### 4.2. Domestic Pigs—Natural and Inbred Resistance

Although the case fatality rate for domestic pigs infected with virulent ASF viruses usually approaches 100 per cent, individual pigs in experimental studies have been reported to survive without developing marked clinical signs of ASF [37,138]. The ability of domestic pigs to not only survive infection with virulent ASF viruses, but to remain healthy and productive is supported by reports of domestic pig populations in endemic areas in Malawi and Mozambique where a higher than expected proportion of healthy pigs have antibodies to ASF virus [36,37,139,140]. An experimental study to determine whether the resistance observed in pigs from northern Mozambique is inherited was performed by challenging the offspring of serologically positive pigs with the two ASF outbreak viruses circulating in the population of origin [141]. The study failed to demonstrate resistance, as only one out of 105 pigs survived challenge [37]. However, a similar study of longer duration in Portugal suggested that repeated inbreeding of pigs that showed resistance to the pathogenic effects of ASF virus resulted in an increased survival rate [142].

Another possible means of attaining resistance involves the introduction of wild suid genetic material into domestic pigs. Anecdotal reports of mating between bushpigs and domestic pigs were explored at the Agricultural Research Council-Onderstepoort Veterinary Institute, South Africa in an unpublished study. A boar, two sows and a litter of weaned piglets purported to be the result of bushpig-pig cross-breeding were obtained for the investigation of resistance. However, genetic analyses indicated that the pigs were *Sus scrofa* crosses of domestic pigs and European wild boars. Likewise, a genetic study of two domestic pig populations in ASF-endemic areas in western Kenya that demonstrate improved ability to survive ASF did not reveal any relatedness to warthogs or bushpigs [143]. Anecdotes of mating between domestic pigs and bushpigs resulting in viable offspring are frequent and generally accepted. The likelihood of successful cross-breeding between the two species is, however, very small as they are genetically distant; the *Sus/Porcula*-*Potamochoerus/Phacochoerus* lineages are estimated to have diverged ~9.9 million years ago [144]. However, what was of relevance in the Kenyan study is that resistance to ASF was higher in the more homogeneous population of local breed pigs than in a genetically heterogeneous population, in spite of the >54 per cent local pig ancestry in the latter population [143]. These findings concur with those of Vasco [142] and suggest that breeding of pigs for resistance to ASF may be a viable, if not largely overlooked, avenue of research. The results further suggest a multifactorial basis to the inheritance of resistance to ASF.

### 4.3. Domestic Pigs—Genetic Engineering

Genetic engineering is an alternative line of investigation that holds promise for delivering results far more rapidly than traditional selective animal breeding approaches. Two genome editing avenues have been explored to date, *viz.* (i) replacement of immune modulatory factors with the wild suid orthologues and (ii) modification of putative virus receptor gene domains. The former approach was used to produce domestic pigs with a warthog RELA orthologue associated with resistance to ASF [145]. However, challenge with a moderately virulent virus did not protect the pigs from developing fatal ASF [146]. In contrast, the virus receptor modification approach, although not successful for ASF [147], was used to breed biologically functional pigs that are fully resistant to porcine reproductive and respiratory syndrome (PPRS) genotypes, using the CRISPR/Cas system to either delete the exon 7 domain of the *CD163* gene [148] or to replace it with the corresponding exon found in the human *CD163-like 1* gene [149]. These successes provided the impetus for attempts to genetically engineer ASF-resistant pigs. However, in contrast to the promising PPRS-resistance results, the CRISPR/Cas gene-edited pigs lacking *CD163*, which is a putative receptor for ASF virus, were shown to be fully susceptible to infection with ASF genotype II Georgia virus [147]. Efforts are ongoing to pursue both the wild suid allele introgression into the pig genome [145] as well as the domestic pig receptor modification/deletion route, despite the setbacks experienced. Although efforts to modify the pig host with genome editing technologies have not proved successful, historical pig breeding results [142] indicate that host-centred approaches to limit the impact of ASF may be a valuable route to pursue in parallel to vaccine development efforts. Unlike expensive gene-editing, traditional selective breeding for ASF resistance is well within the reach of resource-poor farmers in endemic settings, as long as they have access to survivor pigs that are often well-adapted to the local environment, with which to initiate breeding programmes. Recent calls to avoid/re-evaluate mass culling, particularly in low- to middle-income, non-commercial settings [150], if heeded, would ensure that more survivor pigs become available for breeding programmes initiated by local communities; this is currently a constraint. The other major impediment to success is the amount of time needed to achieve the required level of homogeneity for stable inheritance of the resistance traits and the concomitant risk of loss of other favourable traits in the process. This can be countered, to some extent, by establishing multiple breeding lines.

The loss of genetic diversity and uniqueness of local breeds, together with the loss of pigs potentially resistant to ASF, is an often overlooked negative impact of mass culling. When performed to the extreme, as was the case in Haiti from 1978–1982, when all pigs (>384,000) on the island were culled in response to ASF outbreaks, mass culling can result in extinction of local pig breeds, effectively wiping out tens to hundreds of years of adaptation to a specific environment. Mass culling of pigs in Haiti led to the extinction of the Haitian Creole pig breed that was well-suited to local conditions, unlike the Duroc, Yorkshire and Hampshire breeds that were used to restock the island after the ASF cull was concluded [151]. Efforts are now underway to recover the Creole breed. A fuller appreciation for the value of locally-adapted pig breeds is driving efforts to document pig breed diversity, as first step in conserving this valuable and understudied genetic resource, particularly in Africa [152]. These efforts marry well with the need identified here to establish local ASF-resistant pig-breeding initiatives, in tandem with biosecurity improvements, whilst a vaccine is being developed.

## 5. Discussion

As shown in the previous sections, human activities drive ASF transmission, including in the current outbreaks occurring across four continents, and control can be achieved by applying basic biosecurity measures. This has furthermore been known for almost a hundred years and also been more and more widely communicated as the global interest in ASF has increased. Exemplifying this, in Uganda where ASF is endemic, disease prevention and biosecurity have been increasingly targeted in many trainings and development programmes during the last decades, including a manual specifically targeting ASF control in the smallholder pig sector [153]. In response to the increased understanding of the human dimension of ASF epidemiology, several studies have investigated smallholders’ knowledge, awareness, perceptions and practices concerning biosecurity, often referred to as KAP studies (knowledge, attitude, practices) [75,76,79,113,154]. These studies in resource-limited settings generally report that implementation of biosecurity is far from fully executed but come to diverging conclusions about what is hindering implementation [155,156,157,158]. Some report that most smallholders, as well as other stakeholders in the pig production value chain as well as those relating to other species are knowledgeable and aware of existing options for control including biosecurity [115,159], whereas others report knowledge gaps in this regard [155]. These reported differences might to some extent not represent actual differences in participant knowledge, but rather mirror effects of study design and of researchers’ epistemology and views on local and situated knowledge [111].

Few studies have investigated implementation of biosecurity and how this can be improved in the settings discussed here. Chenais et al. [6] discussed these aspects of biosecurity in the European setting, introducing the concept of “hardware and software biosecurity”, where the software part refers to how people are able to follow the advice and habitually and sustainably implement biosecurity routines that are set up. Biosecurity hardware refers to the biosecurity infrastructure in place (buildings, barriers, material). Both hardware and software need to be in place and in sync to achieve implementation and the desired protective effect. In a recent study from Uganda by Dione et al. [112] the effect on smallholders’ knowledge and changes in practices after trainings on biosecurity was investigated. In summary, participants’ knowledge improved after receiving training, but their practices changed to a lower degree. Young et al. [155] described experience from research on smallholder farm biosecurity in Cambodia and Laos, taking a knowledge-based perspective but concurrently concluding that biosecurity needs to be adapted to local contexts to achieve successful implementation. Coffin et al. [160] investigated knowledge and behaviour in the context of an anthrax outbreak in Uganda, concluding that peer pressure, poverty, and lack of health and veterinary infrastructure influenced responses to disease outbreaks more than knowledge. Studies by Chenais et al. [63,115] showed that pig smallholders invest close to nothing in their pigs and in biosecurity but are confident in the capacity of biosecurity to protect pigs from ASF. In one of the studies [63], participants further expressed generally low acceptance for suggested hypothetical biosecurity interventions. The authors concluded that increased implementation of biosecurity requires not only low-cost interventions, but also that measures are developed and adapted in participation with the end-users. A recent study from the Philippines shows that such participatory adaptation can be performed regarding pig management, but that it requires long-term engagements and good local anchorage [161]. In this regard it is important to consider that promoting biosecurity measures that cannot be implemented locally will not only prevent their implementation and thus not serve the intended purpose for disease control but might also reduce general credibility of the agency that is delivering messages that are seen as unrealistic. Traditions and cultural expressions can furthermore vary between communities that are geographically or culturally close and that look homogeneous to the outsider researcher [162]. In the participatory process of local adaptation of biosecurity measures it is furthermore important to consider plurality and power dynamics among participants, including from the same communities, as these aspects will affect how people can act on advice and express opinions [162,163]. In Papua New Guinea, for example, the different roles of men and women in pig rearing as well as cultural differences among different ethnic groups in traditional pig rearing have to be considered [59].

Sustained, increased implementation of biosecurity will furthermore only be achieved if concerned stakeholders appreciate the benefits of the interventions and consider it a priority among all other necessary tasks to perform in the complex day-to-day realities of subsistence pig farming [116,164]. In this regard local people, researchers, governments and multinational organisations might have conflicting agendas regarding priority diseases to control or eradicate [162,165]. The research community, governments and multinational organisations tend to prioritise transboundary diseases with dramatic impact on trade and national economies such as ASF. Animals in subsistence farming systems in resource-poor settings on the other hand often suffer from constant subclinical infections of numerous endemic diseases as well as malnutrition with stunting and low productivity as consequences. Priorities of smallholder farmers are linked to their day-to-day reality, therefore favouring control of endemic disease as well as husbandry and feed improvements rather than eradicating transboundary diseases that might appear only at irregular intervals [111,162,165,166]. In this regard, we believe that the ability of biosecurity to prevent multiple diseases and thus improve the general herd health with one silver bullet is preferred compared to the limited impact of for example a univalent vaccine. However, particularly in newly-infected countries, experience of ASF with heavy losses of pigs may be a strong incentive for improvement of biosecurity.

Recent increased understanding about the human dimension of ASF epidemiology includes identification of behaviours and actions in the smallholder pig value chain critical for disease transmission risks. Adopting a pure veterinary scientist perspective, the sequel to such findings is that ASF control could be achieved if the identified behaviours simply changed or ceased [64,112]. Taking a more holistic and interdisciplinary perspective encompassing the livelihood situation of poor farmers and value chain actors, it becomes evident that ASF control cannot be sustainably achieved without understanding the underlying reasons behind the identified risk behaviours, and also including this dimension in disease control efforts. For example, trying to prevent specific actions such as trading in sick pigs will not be effective if this action is a desperate coping mechanism to evade poverty or obtain cash to pay for school fees. A social network analysis of pig-keeping households in a border area between eastern Uganda and western Kenya indicated that healthy pigs were usually traded locally within a 5 km radius, but pigs sold during outbreaks were usually sold to traders or to farms outside the usual radius, showing that the sellers were aware of possible negative results from the sales of infected pigs [167]. Likewise, educating butchers about slaughter hygiene and biosecurity remains futile if slaughter facilities lack even basic equipment such as running water or possibilities for safe disposal of risk material. The need to provide hygienic slaughter facilities was highlighted in studies commissioned by FAO of the pig sectors in Kenya, Burkina Faso and DRC; the conditions under which pigs were slaughtered were described as ranging from unsatisfactory to deplorable [45,46,168]. Providing improved infrastructure for slaughter of animals is usually a function of government, whether at national or local level, and needs to be taken seriously in order to prevent zoonotic and public health related diseases.

In summary, it seems that the actual implementation of known biosecurity measures in the production systems and in all parts of smallholder pig production value chains is the key to prevention and control, and that challenges other than the strictly technical are hindering this from happening. From recent global outbreaks of other infectious diseases such as Ebola in West Africa, as well as the ASF epidemic in Europe and Asia, it has similarly been observed that technical, biological or epidemiological knowledge are not sufficient to achieve disease control or prevention [63,169,170]. Experience from each of these disparate disease events shows that to prevent transmission and achieve disease control, biosecurity regimens need to be adapted so that they are both scientifically relevant and possible to act on for the people concerned. For this, local disease drivers need to be identified and understood, and livelihood contexts and situated knowledge of stakeholders taken into account [63,111,160]. The understanding of how human behaviour drove transmission, achieved through a multi-disciplinary, biosocial, bottom-up and community-centred approach drawing on social science competence was critical to finally bringing the Ebola epidemic under control [170]. Instead of advancing technical solutions, increasing the understanding of local sociocultural, economic and political dimensions as well as individual keys to effective communication seem to be what paves the pathway to increased implementation of biosecurity [115]. This applies especially to poor pig producers in smallholder systems, the focus populations of this review. Finally, as ASF conquers new countries and reveals both commonalities and differences, especially in smallholder and subsistence pig farming contexts, it becomes increasingly evident that management of ASF cannot be achieved through conventional approaches alone. A wide range of options should be explored, using a multi-disciplinary approach to ensure that disease management is compatible with livelihoods, cultures and biodiversity conservation.

## Figures and Tables

**Table 1 vaccines-09-00116-t001:** Determinants of biosecurity measures for African swine fever.

Source and Transmission	Importance	References
Direct contact transmission	Important	[81,96]
Ingestion of infected material due to persistence in tissues, meat	Important	[81,83]
Contact with contaminated fomites	Moderately important	[71,77]
Persistence in the environment outside host/host tissues	Less important	[81,97,98]
Presence of African wild suids	Important locally	[96]
Presence of wild/feral *Sus scrofa*	Important regionally	[63,71]
Biological tick vector	Important locally	[96]
Other blood-sucking arthropods (only stable flies (*Stomoxys calcitrans*) have been shown to be competent mechanical vectors of the virus)	Might play a role locally (no importance demonstrated under natural conditions)	[99,100,101]

**Table 2 vaccines-09-00116-t002:** Biosecurity measures recommended to prevent common transmission routes based on the authors’ experience and knowledge of the disease transmission.

Source and Transmission	Preventive Measures
Direct contact with infected pigs	Confine pigs in pig-proof pens Acquire new pigs only from known safe sources Quarantine and observe new pigs for at least 15 days Separate any pigs showing clinical signs
Ingestion of infected material	Do not feed swill containing meat Heat swill to destroy the virus Do not allow pigs to scavenge (confine pigs in pig-proof pens) Safe disposal of infected material (carcasses, slaughter waste)
Contact with fomites	Limit access to the pigs (carers and health service providers only) Provide a change of footwear Disinfectant footbaths (effective product and brush for cleaning) Do not share equipment or clean thoroughly and disinfect before use Do not accept leftover feed or bedding from producers whose pigs have died Check vegetation supplied as feed for visible signs of contamination
Biological tick vector from warthogs	Confine pigs in pig-proof premises (to keep pigs in and warthogs out)
Biological tick vector in domestic pigs	House pigs in concrete pens with smooth finish
Stable flies	Remove breeding places (grass cuttings, discarded bedding) Use commercial fly control products

**Table 3 vaccines-09-00116-t003:** Constraints identified for implementation of recommended basic biosecurity measures based on literature cited in the text and personal experience of the authors.

Recommended Biosecurity Measure	Identified Constraints
Permanent confinement of pigs in pig-proof pens	Cost and labour of providing housing, feed and water Socio-cultural norms and practices
Purchase of pigs from known safe sources	Lack of information about pigs sold through agents and live markets Lack of choices for sourcing pigs
Provision of safe feed	Commercial rations not available or affordable Tradition of keeping pigs as a natural and ecological converter of food waste into a valuable asset Swill consisting of leftover catering waste is the most affordable feed Heating swill before feeding is expensive in terms of the fuel used
Safe disposal of pig carcasses and offal	Lack of waste disposal services Resistance to disposing of meat considered edible Cost and labour for burning or burying Socio-cultural norms and practices
Restricting access to pigs	Sale through agents may be the only option Socio-cultural norms and practices
Use of disinfection and protective clothing	Lack of funds to purchase disinfectants and protective clothing Disinfectants may not be available Not possible if pigs are not confined

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
