# Peer review of "With or without a Vaccine—A Review of Complementary and Alternative Approaches to Managing African Swine Fever in Resource-Constrained Smallholder Settings"

_vaccines, 2021, doi:10.3390/vaccines9020116_

Round 1

Reviewer 1 Report

This is an interesting and well researched review which explores different options for control of African swine fever in pigs within the small holder sector in different countries. The review highlights the key biosecurity measures that could be introduced and issues with their introduction in different settings. Social and cultural differences which may come into play in different countries are also described and examples given. An alternative strategy, breeding or gene editing to develop pigs with genetic resistance to disease is also considered. This section is over-optimistic about the prospects for introducing ASFV resistant domestic pigs in general and in a small holder setting in particular.

Specific points:

  1. On line 386 the authors discuss evidence for genetic resistance of wild African suids and some domestic pigs showing resistance to ASFV as opposed to tolerating the infection. The authors should define the meaning of these terms and discuss in more detail the evidence for resistance rather than tolerance.
  2. The authors should discuss some of the difficulties in breeding for disease resistance in pigs. They should also discuss the difficulties that would be encountered in producing gene-edited pigs that are resistant to ASFV and introducing this trait, most likely from a commercial breed, into the diverse pig population found in many small holder settings.

Attempts to produce gene-edited pigs resistant to ASFV have been very long shots with little scientific evidence to back up the approach taken.

Author Response

This is an interesting and well researched review which explores different options for control of African swine fever in pigs within the small holder sector in different countries. The review highlights the key biosecurity measures that could be introduced and issues with their introduction in different settings. Social and cultural differences which may come into play in different countries are also described and examples given. An alternative strategy, breeding or gene editing to develop pigs with genetic resistance to disease is also considered. This section is over-optimistic about the prospects for introducing ASFV resistant domestic pigs in general and in a small holder setting in particular.

Thank you for your review and positive feedback.

Specific points:

  1. On line 386 the authors discuss evidence for genetic resistance of wild African suids and some domestic pigs showing resistance to ASFV as opposed to tolerating the infection. The authors should define the meaning of these terms and discuss in more detail the evidence for resistance rather than tolerance.

Thank you for highlighting the need to expand on and differentiate between what is meant by resistance and what is meant by tolerance. The relevant section now reads: “For the purposes of this review an ASF-resistant animal is defined as one that is able to limit pathogen burden such that no negative health impacts are observed” whereas an ASF-tolerant animal is one the displays negative health impacts but does not succumb to the disease”.

  1. The authors should discuss some of the difficulties in breeding for disease resistance in pigs. They should also discuss the difficulties that would be encountered in producing gene-edited pigs that are resistant to ASFV and introducing this trait, most likely from a commercial breed, into the diverse pig population found in many small holder settings.

Thank you for the highlighting the need to clarify that gene-edited pigs are unlikely to make their way into small-holder settings in the foreseeable future. We have indicated the cost constraints and agree that, if gene-edited pigs were to become a viable option in future that they would not be well-suited to small-holder settings. In addressing this concern we have attempted to more clearly define the limitations of gene-editing pigs and to emphasize the value of focussing on local pig breeding efforts to achieve improved levels of resistance, whilst also mentioning constraints.

The greatest impediments to breeding for resistance is that (i) culling severely reduces the pool of starter pigs with which to breed, and (ii) that the levels of inbreeding over multiple generations required to achieve a discernible shift in the right direction may result in the loss of other favourable traits. These challenges have been mentioned.

Attempts to produce gene-edited pigs resistant to ASFV have been very long shots with little scientific evidence to back up the approach taken.

We agree that efforts to produce ASF-resistant pigs through gene editing are in their infancy and not well-suited to resource-constrained settings. However, indicators of success with other diseases suggest that with a better understanding of the basis of ASF resistance that, in future, this may become a viable means of producing ASF-resistant pigs, initially for high-production systems. What we were attempting to convey by including gene editing, is that by broadening research efforts to include host aspects instead of focussing primarily on the virus/vaccine that greater progress can be made with managing ASF. We hope that the changes made to the manuscript bring this point across more clearly.

Reviewer 2 Report

Thank you very much for your review of ‘alternative’ approaches. The question is, alternative approaches to what? Biosecurity is practiced all over the world in relation to ASF and in all pig production systems. Thus, biosecurity is not an alternative approach.

Furthermore, I would suggest to focus the paper on Africa only, as most of the literature cited in the review is focused on Africa. It would be feasible to add the other countries/regions in the discussion. In its current form, it seems that the review is mainly based on information from Africa, where the information is good and plentiful. But the situation of the smallholders in other regions of the world is not as detailed described as the one from Africa. Often it is not clear on which region the paragraphs or sentences are focused on.

Furthermore, the discussion seems to focus on participatory methods, but there is hardly anything mentioned about this in the main text of the review. Either delete large parts of the discussion or add further references on participation into the main text

Some references are cited, which I do not believe contain the issue specified in your text (some are detailed in the specific comments).

Tables 2 and 3 detail biosecurity measures, but it is not clear on what basis these were grouped – there are no references given.

In some parts of the manuscript the author guidelines are still included – not sure, if this was a lack of diligence or a technical issue.

in the title you talk about resource poor settings, this is not coming out in the manuscript.

Specific comments:

Very often the references are quoted: ([5] – eg line 37

Pages 8 and 9 are more a personal discussion than a review

Page 1, line 18: The only option currently available to prevent ASF is strict biosecurity. – you mean in domestic pigs? Would biosecurity help to prevent ASF in wild boar or wild suids?

Page 2, line 54: this should read elimination, not eradication (elimination – which you use correctly in the section 4.1. is different from eradication)

Page 2, line 65: reference 25 is not describing that Belgium or Czech Republic are free from ASF. If you use reference 26 for Belgium, why not use the same source for the Czech Republic?

Page 2, line 75/76: These regions: do you mean the Asia-Pacific region AND Sardinia? Or do you mean THIS region?

Page 2, line 83: you mention here innate resistance as an option for wild boar. More discussion (e.g. in the discussion section) would be needed to substantiate this claim.

Page 3: lines 114 ff: this seems to be some parts of the author guidelines

Page 3, line 137: “while supporting the principle”: which one?

Page 4, line 151: this stratum of the pig population .. in which region? Africa/Asia/Europe?

Page 4, lines 162: are backyard pig farms not also included in production systems with free-ranging pigs?

Page 5, line 199: high loads of ASF virus in excretions. Is this still valid? There are numerous references that detail, that the virus load is much lower in excretions than in blood. And that excretions are not as important as blood.

Page 5, line 200-203: this seems to be text phrase from the author guidelines.

Page 5, lines 205: the only published information on durability: there are many more references on meat (even without processed food). These should be added. E.g. Blome and Dietze 2011, McKercher et al., 1987,

Page 5, line 210: reference 92 is a review, not the original literature on the R0

Page 5, line 215-217: not sure if this sentence belongs into the aforementioned context

Page 5, lines 219 ff: this is about large-scale commercial farms – which you did not want to write about (as mentioned further up).

Page 5, line 225: “not specifically required” – required not at all, or specifically for ASF or for other diseases?

Page 6, line 237: warthogs, domestic pigs and ticks

Page 6, line 238: Pig-associated (which pigs? Domestic, warthogs?

Page 6, line 240: ticks .. are not a concern. (please give a reference)

Page 6, line 251: long distance dispersal of flies? , but with much shorter distances for unfed and partially fed flies – than for fed flies?

Page 6, table 2: please add references, as you did in table 1

Page 7, table 3: please add references, as you did in table 1 – it is not clear if these are your opinions or if they are based on references.

Page 7: from here onwards, the focus is very much on Africa, which is fine, but I would focus the whole manuscript on Africa, then it is more coherent and clear.

Page 8, line 281: some cultures or communities: references missing

Page 8, lines 286: please add references

Page 8, lines 301/302: there are other references for feeding ASFV to pigs, not only the first description of Montgomery

Page 8, line 307: not sure if the reference 118 is correct in this context.

Page 8, lines 324: please add references – this part of the manuscript it seems more like an opinion or a discussion, not like the parts before, where references are cited.

Page 9, lines 338ff: is this practical? Please give also a reference for this part.

Page 9, lines 340ff: scavenging pigs are outside, and most of the time you do not check the environment for carcasses to dispose of them.

Page 9, line 348: please add reference

Page 9, lines 351: what is wrong with not accepting remains as feed for pigs in situations, where people never felt hunger?

Page 9, line 362/363: improving diagnostic capabilities of farmers. What is meant by this?

Page 10, line 393: there are additional references detailing this

Page 10, lines 405 ff: the origin of the piglets is not clear. Is it important for the review to detail, that the piglets were purported as being wild, but they were not? This part could be shortened.

Page 10, line 426: it would be good to state what Vasco et al have reported

Discussion:

Page 11, line 471: current outbreaks where? In which region? Thus it would be good to focus the paper only on Africa.

Page 11, line 484: delete the space between ‘species’ and ‘are’

Page 13, line 578: simply changed or ceased – please give a reference for this

Page 13, line 573: if you mention Covid 19 (presumably in order to use the hype of the moment) then please also add a reference for it

Page 13, line 585: “the focus populations of this review”. It is not clear what you mean with the reference 113? Why do you add reference 113 if you talk about the current review?

Page 13, line 589: which may appear revolutionary: which options appear revolutionary?

Author Response

Thank you for a very comprehensive review.

Responses to general comments

Thank you very much for your review of ‘alternative’ approaches. The question is, alternative approaches to what? Biosecurity is practiced all over the world in relation to ASF and in all pig production systems. Thus, biosecurity is not an alternative approach.

While we agree that biosecurity is practiced throughout the world in the modern pig farming sector, in the sector we are considering, mostly no measures to manage ASF effectively are in place. The problem furthermore is that the level of biosecurity recommended for modern, technically advanced pig farms is neither feasible nor necessary for smallholder farmers to keep their pigs safe from ASF. While in some of the EU member states smallholder farmers who cannot comply with the exacting biosecurity standards implemented on industrialized pig farmers have been forced to stop pig farming, this could hardly be applied to the millions of poor smallholder pig farmers in developing countries worldwide who depend on their pigs for various reasons, including as a very important source of income. We are therefore proposing a specific basic level of biosecurity as an alternative to (1) no management, (2) comprehensive and efficient vaccination to protect the pigs despite a lack of biosecurity and (3) enforcing a ‘’one size fits all’’ approach to biosecurity in all types of pig farms.  Please note also that we refer to ‘’alternative and complementary’’ measures as we would agree with anyone who says that vaccination alone will not protect fully against ASF.

Furthermore, I would suggest to focus the paper on Africa only, as most of the literature cited in the review is focused on Africa. It would be feasible to add the other countries/regions in the discussion. In its current form, it seems that the review is mainly based on information from Africa, where the information is good and plentiful. But the situation of the smallholders in other regions of the world is not as detailed described as the one from Africa. Often it is not clear on which region the paragraphs or sentences are focused on.

We believe that what we are proposing would be suitable in a wide range of smallholder pig farms wherever they are as the knowledge and experience of the co-authors does extend beyond Africa, and we do not want this paper to be written off as just another paper about African problems that are not relevant elsewhere. However, we have added a sentence to explain why the focus and examples given are largely from Africa because ASF has been endemic in Africa for very much longer than on the other continents now affected.

Furthermore, the discussion seems to focus on participatory methods, but there is hardly anything mentioned about this in the main text of the review. Either delete large parts of the discussion or add further references on participation into the main text.

The main text of the review presents the available biosecurity measures required to prevent ASF in smallholder pig farms and constraints for their implementation. The use of participatory methods applies to the entire spectrum of measures and it would have been tedious to introduce the concept into each of the sections that deal with a specific aspect of biosecurity. For many small-scale pig farmers education and training have proven sufficient to result in improved biosecurity, but in some contexts where this is not the case, particularly where the constraints are socio-cultural and not just financial or due to lack of information. Some additions and changes to make this clearer have been added on lines 202-03, 240, 270-72..

Some references are cited, which I do not believe contain the issue specified in your text (some are detailed in the specific comments).

This has been addressed or explained in the specific comments. If there are any that are not detailed in the specific comments we apologise but we believe that our use of references is appropriate, which we have demonstrated in our specific responses.

Tables 2 and 3 detail biosecurity measures, but it is not clear on what basis these were grouped – there are no references given.

The grouping of the measures in Tables 2 and 3 is based on the knowledge and experience of the authors. All of them are covered in the text and it does not seem necessary to add the references to the tables as well.

In some parts of the manuscript the author guidelines are still included – not sure, if this was a lack of diligence or a technical issue.

Parts of the author guidelines are included in the text in the final review version the publisher sent us to work from – they were not visible in the version we submitted so it must be a technical issue. This is the first time that the corresponding author has been asked to use a template from the publisher and it may contain some hidden text that reappeared in the review version, which also receives attention from the publisher. This technical issue has been addressed by deletion of the offending text.

in the title you talk about resource poor settings, this is not coming out in the manuscript.

Just about all the settings we write about are resource poor and I think it does come out in the paper as it is mentioned in both the introduction and the section on free-range pig-farming, and the fact that it is in the title should in fact be sufficient to inform the reader as to our target context.

Specific comments

Very often the references are quoted: ([5] – eg line 37

Thank you for picking up on the extra bracket, we thought we had found all of them. We have checked again with the search function and they have now all been eliminated.

Pages 8 and 9 are more a personal discussion than a review

Re pages 8 and 9, discussion of the topics is necessary as they are important in the context, and references have been used when available, but we did not see this as purely a review of literature and we have included our own experience and opinions.

Page 1, line 18: The only option currently available to prevent ASF is strict biosecurity. – you mean in domestic pigs? Would biosecurity help to prevent ASF in wild boar or wild suids?

Thank you for drawing our attention to this. The preceding sentence refers to both wild boars and domestic pigs. Biosecurity is appropriate for both. To avoid misunderstanding, ‘wild pigs’ in the preceding sentence has been changed to ‘wild boars’ because the concept does not include African wild suids. The prompt finding and removal of wild boar carcasses is definitely a biosecurity measure, as is the ban on feeding, which encourages wild boars to congregate. Unfortunately the restriction on the number of words permitted for the Abstract precludes further clarification at this point.

Page 2, line 54: this should read elimination, not eradication (elimination – which you use correctly in the section 4.1. is different from eradication)

Changed to elimination.

Page 2, line 65: reference 25 is not describing that Belgium or Czech Republic are free from ASF. If you use reference 26 for Belgium, why not use the same source for the Czech Republic?

Marcon et al. (2020) (reference 25) describes Czech Republic as being free from ASF (page 2 line 4 from the top: ‘at the time of writing, only Czech Republic has been able to eradicate ASF in the wild, whereas the other EU countries are still facing the presence of the virus in their wild boar populations..’). However, for consistency we have replaced it with the Czech Republic declaration of freedom, which is more appropriate. Reference 26 is the Belgian declaration of freedom in the wild boar population so we cannot use it for Czech Republic.

Page 2, line 75/76: These regions: do you mean the Asia-Pacific region AND Sardinia? Or do you mean THIS region?

Changed to ‘this region’ because eradication seems to be fairly advanced in Sardinia and they would be unlikely to consider vaccination even if it becomes available. We may have used ‘these regions’ because in fact Asia-Pacific represents two continents (Asia and Oceania) but maybe it is fair enough to consider it to be one region for the sake of clarity.

Page 2, line 83: you mention here innate resistance as an option for wild boar. More discussion (e.g. in the discussion section) would be needed to substantiate this claim.

The whole issue is discussed in more detail in Section 4.

Page 3: lines 114 ff: this seems to be some parts of the author guidelines

The text, which to the best of our knowledge was not visible there when we submitted the manuscript, has been deleted.

Page 3, line 137: “while supporting the principle”: which one?

To ensure clarity ‘of fencing’ has been added.

Page 4, line 151: this stratum of the pig population .. in which region? Africa/Asia/Europe?

It would be applicable to that pig population wherever it occurs, including in Latin America and the Caribbean if the disease should decide to cross the Atlantic again; to clarify this, ‘wherever it occurs can’ has been added.

Page 4, lines 162: are backyard pig farms not also included in production systems with free-ranging pigs?

Sometimes they may be, but many or most are not because the pigs are permanently enclosed.

Page 5, line 199: high loads of ASF virus in excretions. Is this still valid? There are numerous references that detail, that the virus load is much lower in excretions than in blood. And that excretions are not as important as blood.

We are aware that blood contains the highest amounts of virus but none of those references suggest that infective amounts of virus are not present in other secretions and excretions as well. If that were the case, contact infection experiments e.g. that of Eblé et al (2019) would not be successful as there is no evidence that blood was involved in the experiments. Other excretions may not be as ‘important’ as blood but if they were unimportant in transmission there would be far less ASF about. The sentence has been changed to be clearer on this point.

Page 5, line 200-203: this seems to be text phrase from the author guidelines.

Deleted.

Page 5, lines 205: the only published information on durability: there are many more references on meat (even without processed food). These should be added. E.g. Blome and Dietze 2011, McKercher et al., 1987

To our knowledge there are not ‘many’ more references on pork (perhaps on other meat) reporting primary research to determine the durability of ASFV in pork. We did not include reviews and risk analyses that refer to published work. McKercher et al (1987) is included and refers to Parma ham. We could perhaps have included McKercher et al (1978) but that refers to partly cooked canned hams and dried pepperoni and salami sausages, but no virus was obtained from the cooked product and neither the dried pepperoni nor salami sausages yielded virus after the required curing period, so the results do not add very much to the information provided.

An exhaustive search of the Web of Science, Scopus and Pubmed databases as well as Google scholar, failed to find any relevant publication (or any publication) by Blome & Dietze 2011, but if the reviewer could supply us with a copy we would be delighted to include it if it contains different information.

Page 5, line 210: reference 92 is a review, not the original literature on the R0

Certainly, but we considered the information to be relevant. Reviews can provide convenient summaries and analyses of primary information, as this one does; we use it as a reference to the fact that the spread of ASF can be either fast or slow, without having to cite a string of references on R0 as that is not the main topic of this review.

Page 5, line 215-217: not sure if this sentence belongs into the aforementioned context

Thank you for pointing this out. We have expanded it slightly and moved it to a separate paragraph, as it is not specific to a particular transmission route but is often the determinant of how the virus spreads regardless of the transmission route.

Page 5, lines 219 ff: this is about large-scale commercial farms – which you did not want to write about (as mentioned further up).

It seemed appropriate to expand a little on why we did not include an in-depth discussion of biosecurity on large-scale commercial farms. We do not see any reason to remove this paragraph.

Page 5, line 225: “not specifically required” – required not at all, or specifically for ASF or for other diseases?

As the sentence states, not specifically required to prevent ASF.

Page 6, line 237: warthogs, domestic pigs and ticks

Ticks have been added, thank you.

Page 6, line 238: Pig-associated (which pigs? Domestic, warthogs?

Sentence altered to be clearer, thank you.

Page 6, line 240: ticks .. are not a concern. (please give a reference)

The sentence has been changed to ‘not known to be a concern’. That is closer to the truth and does not need to be referenced – thank you for pointing it out.

Page 6, line 251: long distance dispersal of flies? , but with much shorter distances for unfed and partially fed flies – than for fed flies?

Presumably, but the authors did not specify that so for us to do so would not be accurate citation.

Page 6, table 2: please add references, as you did in table 1 and Page 7, table 3: please add references, as you did in table 1 – it is not clear if these are your opinions or if they are based on references.

References have not been added for the reasons provided in the general responses. We cannot attribute the recommendations to any particular reference, they are very widely reported and discussed. We did try adding references to table 3 but decided to remove them for that reason.

Page 7: from here onwards, the focus is very much on Africa, which is fine, but I would focus the whole manuscript on Africa, then it is more coherent and clear.

We have indicated why we prefer not to imply that this paper is only relevant for Africa, because it was largely inspired by the fact that whether outbreaks of ASF occur in Africa, Asia, the Pacific region or Eastern Europe, there is a predominance of backyard farms and village pigs among the epidemiological units affected (as reflected in the reports to OIE). We believe that the review is not relevant only to Africa, and that other regions can benefit from the long African experience of ASF. If we restrict it to Africa it may be written off as something unique to Africa and irrelevant anywhere else; if that was the case we would not have written it.

Page 8, line 281: some cultures or communities: references missing

References added.

Page 8, lines 286: please add references

The example given is described in the reference cited and is unique to that reference, so we cannot add more references. We have checked the sentence again and it seems to be clear that we are citing a reference about an initiative in Timor-Leste.

Page 8, lines 301/302: there are other references for feeding ASFV to pigs, not only the first description of Montgomery

Yes, but to provide a whole string of references for something as well-known as that seems unnecessary, and Montgomery was the first person (and one of the few) to undertake painstaking experiments with a range of foodstuffs fed in different forms to pigs and his findings have never been disproven, so we are happy to give him credit in the anniversary year of his providing the very first publication on ASF.

Page 8, line 307: not sure if the reference 118 is correct in this context.

It is correct and appropriate.

Page 8, lines 324: please add references – this part of the manuscript it seems more like an opinion or a discussion, not like the parts before, where references are cited.

Line 324 includes the two relevant references, so presumably the reviewer is referring to the next paragraph (probably the additions made to the text so far have added a line). Much of what follows here is discursive, as there is not a lot of peer-reviewed literature on the subject, and most of it comes from the authors’ experience, which we think is legitimate to reflect in a review. Some of the information may come from mission reports by co-authors that are confidential so we cannot very well cite them although the information itself is not protected.

Page 9, lines 338ff: is this practical? Please give also a reference for this part.

It seems to be a fairly harmless recommendation and is practical in the sense that contamination of crops or vegetation with blood or faeces can be highly visible if the material is fresh enough to pose a threat. Reference 73 mentions this as a source of infection but makes no recommendations for avoiding the problem and the first author mentioned in a meeting that one backyard farmer had admitted to collecting grass from a wild boar deathbed site and feeding it to his pigs, although he saw that it was contaminated. We cannot share other peoples’ stories that they have not chosen to publish themselves but we believe we may take note of them and base upon them what we feel may be appropriate recommendations.

Page 9, lines 340ff: scavenging pigs are outside, and most of the time you do not check the environment for carcasses to dispose of them.

That is so, and that is why we recommend as far as possible not to carelessly dispose of the carcasses of pigs. Obviously if wild boars are an issue there are several more reasons not to allow pigs to roam freely so this paragraph refers to domestic pigs in areas where the most likely carcass will be that of another pig. There are, incidentally, no documented cases of environmental contamination by the carcasses of African wild suids, which usually do not contain significant amounts of ASF virus.  

Page 9, line 348: please add reference

Another reference to unpublished data of one of the authors has been added.

Page 9, lines 351: what is wrong with not accepting remains as feed for pigs in situations, where people never felt hunger?

There is nothing wrong with that but if the pig died of ASF the uncooked remains should not be fed to pigs due to the probability of infection. The comment about people who have not felt hunger refers to the fact that even if they are horrified by people cooking and eating dead animals including those that have died of ASF it is not within their rights to disapprove of the practice. It is a bit like disapproving of the cannibalism admitted to after the Andean plane crash – none of us know what we might do to survive under similar circumstances.

Page 9, line 362/363: improving diagnostic capabilities of farmers. What is meant by this?

Enabling farmers to recognise the signs of ASF to support early detection and reporting of outbreaks. It often achieves this, but can also prompt the panic selling of pigs. The sentence containing the phrase has been split for the sake of clarity.

Page 10, line 393: there are additional references detailing this

Agreed, but there are so many references already that we tried to be selective, because this is not a systematic literature review and its main focus is not on pigs surviving individual infection under experimental conditions but on pigs surviving infection with virulent viruses under natural conditions.

Page 10, lines 405 ff: the origin of the piglets is not clear. Is it important for the review to detail, that the piglets were purported as being wild, but they were not? This part could be shortened.

The section has been shortened accordingly to remove unnecessary information.

The genetic evidence showed very clearly that the piglets were pure Sus scrofa and therefore there was no evidence of genetic inputs from Potamochoerus larvatus, which differs genetically from domestic pigs and wild boars. Further investigation of the farms where the animals were bought revealed that they were Eurasian wild boar-domestic pig crosses, but the local farmers called the wild boars by the same common name as they would use for bushpigs (in fact a direct translation into their language of ‘bush pig’). To explain all of that would make the paragraph longer, but we think that what is written even in the new shorter version makes the situation clear – we do know what the origin of the pigs was and we did not purport that they were wild, only that they were purported to result from hybridization with wild pigs (bushpigs), which turned out not to be the case.

Page 10, line 426: it would be good to state what Vasco et al have reported

As this is stated in the previous paragraph, i.e. that inbred pigs, i.e. pigs that were more genetically homogeneous showed more resistance, we did not think it necessary to repeat it.

Discussion:

Page 11, line 471: current outbreaks where? In which region? Thus it would be good to focus the paper only on Africa.

The sentence has been modified accordingly.

Page 11, line 484: delete the space between ‘species’ and ‘are’

Space deleted, thank you.

Page 13, line 578: simply changed or ceased – please give a reference for this

References added.

Page 13, line 573: if you mention Covid 19 (presumably in order to use the hype of the moment) then please also add a reference for it

The reference to the COVID-19 pandemic had nothing to do with the hype of the moment, it simply in some countries including the country of two of the authors has provided magnificent examples of how human behaviour – or misbehaviour – can feed a pandemic. However, combing the plethora of literature that has emerged to find something that may not be there, but is simply on our television screens every day, is beyond the scope of this review and we have deleted it.

Page 13, line 585: “the focus populations of this review”. It is not clear what you mean with the reference 113? Why do you add reference 113 if you talk about the current review?

The citation has been moved to the previous sentence, where we agree with you that it is more applicable, thank you.

Page 13, line 589: which may appear revolutionary: which options appear revolutionary?

Recommending that people who in any case are consuming their pigs that become victims of ASF should cook them well may be considered revolutionary and so may genetically modified pigs, but on mature reflection we believe it better not to emphasise that aspect too much, thank you for assisting the thought processes on that, and for a thorough and thoughtful review.

Round 2

Reviewer 2 Report

Responses to general comments

Thank you very much for your review of ‘alternative’ approaches. The question is, alternative approaches to what? Biosecurity is practiced all over the world in relation to ASF and in all pig production systems. Thus, biosecurity is not an alternative approach.

While we agree that biosecurity is practiced throughout the world in the modern pig farming sector, in the sector we are considering, mostly no measures to manage ASF effectively are in place. The problem furthermore is that the level of biosecurity recommended for modern, technically advanced pig farms is neither feasible nor necessary for smallholder farmers to keep their pigs safe from ASF. While in some of the EU member states smallholder farmers who cannot comply with the exacting biosecurity standards implemented on industrialized pig farmers have been forced to stop pig farming, this could hardly be applied to the millions of poor smallholder pig farmers in developing countries worldwide who depend on their pigs for various reasons, including as a very important source of income. We are therefore proposing a specific basic level of biosecurity as an alternative to (1) no management, (2) comprehensive and efficient vaccination to protect the pigs despite a lack of biosecurity and (3) enforcing a ‘’one size fits all’’ approach to biosecurity in all types of pig farms. Please note also that we refer to ‘’alternative and complementary’’ measures as we would agree with anyone who says that vaccination alone will not protect fully against ASF. 

Thank you very much for your answer. It would be good to include this reasoning into the manuscript

Furthermore, I would suggest to focus the paper on Africa only, as most of the literature cited in the review is focused on Africa. It would be feasible to add the other countries/regions in the discussion. In its current form, it seems that the review is mainly based on information from Africa, where the information is good and plentiful. But the situation of the smallholders in other regions of the world is not as detailed described as the one from Africa. Often it is not clear on which region the paragraphs or sentences are focused on.

We believe that what we are proposing would be suitable in a wide range of smallholder pig farms wherever they are as the knowledge and experience of the co-authors does extend beyond Africa, and we do not want this paper to be written off as just another paper about African problems that are not relevant elsewhere. However, we have added a sentence to explain why the focus and examples given are largely from Africa because ASF has been endemic in Africa for very much longer than on the other continents now affected.

Thank you very much for your answer. I fully agree that any country can learn a lot from the African experience, and most experience is coming from Africa – please include your reasoning into the manuscript and detail the limitations of having less data on other regions.

Furthermore, the discussion seems to focus on participatory methods, but there is hardly anything mentioned about this in the main text of the review. Either delete large parts of the discussion or add further references on participation into the main text.

The main text of the review presents the available biosecurity measures required to prevent ASF in smallholder pig farms and constraints for their implementation. The use of participatory methods applies to the entire spectrum of measures and it would have been tedious to introduce the concept into each of the sections that deal with a specific aspect of biosecurity. For many small-scale pig farmers education and training have proven sufficient to result in improved biosecurity, but in some contexts where this is not the case, particularly where the constraints are socio-cultural and not just financial or due to lack of information. Some additions and changes to make this clearer have been added on lines 202-03, 240, 270-72..

Not sure your answer: “it would have been tedious to introduce the concept..” is convincing. I will leave this up to the editor.

Some references are cited, which I do not believe contain the issue specified in your text (some are detailed in the specific comments).

This has been addressed or explained in the specific comments. If there are any that are not detailed in the specific comments we apologise but we believe that our use of references is appropriate, which we have demonstrated in our specific responses. 

Ok accepted

Tables 2 and 3 detail biosecurity measures, but it is not clear on what basis these were grouped – there are no references given.

The grouping of the measures in Tables 2 and 3 is based on the knowledge and experience of the authors. All of them are covered in the text and it does not seem necessary to add the references to the tables as well. 

Please detail the fact, that the knowledge and experience of the authors is the basis for grouping the measures in the manuscript.

In some parts of the manuscript the author guidelines are still included – not sure, if this was a lack of diligence or a technical issue.

Parts of the author guidelines are included in the text in the final review version the publisher sent us to work from – they were not visible in the version we submitted so it must be a technical issue. This is the first time that the corresponding author has been asked to use a template from the publisher and it may contain some hidden text that reappeared in the review version, which also receives attention from the publisher. This technical issue has been addressed by deletion of the offending text. 

in the title you talk about resource poor settings, this is not coming out in the manuscript.

Just about all the settings we write about are resource poor and I think it does come out in the paper as it is mentioned in both the introduction and the section on free-range pig-farming, and the fact that it is in the title should in fact be sufficient to inform the reader as to our target context.

Yes, you as the authors know that all the settings are about resource poor situations, but the reader needs more clarification in this.

Specific comments

Very often the references are quoted: ([5] – eg line 37

Thank you for picking up on the extra bracket, we thought we had found all of them. We have checked again with the search function and they have now all been eliminated.

Pages 8 and 9 are more a personal discussion than a review

Re pages 8 and 9, discussion of the topics is necessary as they are important in the context, and references have been used when available, but we did not see this as purely a review of literature and we have included our own experience and opinions. 

That is fine, if you state is as such, as own experience and opinion!

Page 1, line 18: The only option currently available to prevent ASF is strict biosecurity. – you mean in domestic pigs? Would biosecurity help to prevent ASF in wild boar or wild suids?

Thank you for drawing our attention to this. The preceding sentence refers to both wild boars and domestic pigs. Biosecurity is appropriate for both. To avoid misunderstanding, ‘wild pigs’ in the preceding sentence has been changed to ‘wild boars’ because the concept does not include African wild suids. The prompt finding and removal of wild boar carcasses is definitely a biosecurity measure, as is the ban on feeding, which encourages wild boars to congregate. Unfortunately the restriction on the number of words permitted for the Abstract precludes further clarification at this point. 

accepted

Page 2, line 54: this should read elimination, not eradication (elimination – which you use correctly in the section 4.1. is different from eradication)

Changed to elimination.

Page 2, line 65: reference 25 is not describing that Belgium or Czech Republic are free from ASF. If you use reference 26 for Belgium, why not use the same source for the Czech Republic?

Marcon et al. (2020) (reference 25) describes Czech Republic as being free from ASF (page 2 line 4 from the top: ‘at the time of writing, only Czech Republic has been able to eradicate ASF in the wild, whereas the other EU countries are still facing the presence of the virus in their wild boar populations..’). However, for consistency we have replaced it with the Czech Republic declaration of freedom, which is more appropriate. Reference 26 is the Belgian declaration of freedom in the wild boar population so we cannot use it for Czech Republic. 

Thank you very much for including the declaration of freedom of Belgium and the one for Czech Republic.

Page 2, line 75/76: These regions: do you mean the Asia-Pacific region AND Sardinia? Or do you mean THIS region?

Changed to ‘this region’ because eradication seems to be fairly advanced in Sardinia and they would be unlikely to consider vaccination even if it becomes available. We may have used ‘these regions’ because in fact Asia-Pacific represents two continents (Asia and Oceania) but maybe it is fair enough to consider it to be one region for the sake of clarity.

Fine, if you mean it like this.

Page 2, line 83: you mention here innate resistance as an option for wild boar. More discussion (e.g. in the discussion section) would be needed to substantiate this claim.

The whole issue is discussed in more detail in Section 4.

Thank you very much

Page 3: lines 114 ff: this seems to be some parts of the author guidelines

The text, which to the best of our knowledge was not visible there when we submitted the manuscript, has been deleted.

Page 3, line 137: “while supporting the principle”: which one?

To ensure clarity ‘of fencing’ has been added.

Page 4, line 151: this stratum of the pig population .. in which region? Africa/Asia/Europe?

It would be applicable to that pig population wherever it occurs, including in Latin America and the Caribbean if the disease should decide to cross the Atlantic again; to clarify this, ‘wherever it occurs can’ has been added. 

Thank you

Page 4, lines 162: are backyard pig farms not also included in production systems with free-ranging pigs?

Sometimes they may be, but many or most are not because the pigs are permanently enclosed. 

It would be good to include this into the manuscript

Page 5, line 199: high loads of ASF virus in excretions. Is this still valid? There are numerous references that detail, that the virus load is much lower in excretions than in blood. And that excretions are not as important as blood.

We are aware that blood contains the highest amounts of virus but none of those references suggest that infective amounts of virus are not present in other secretions and excretions as well. If that were the case, contact infection experiments e.g. that of Eblé et al (2019) would not be successful as there is no evidence that blood was involved in the experiments. Other excretions may not be as ‘important’ as blood but if they were unimportant in transmission there would be far less ASF about. The sentence has been changed to be clearer on this point.

Thank you very much

Page 5, line 200-203: this seems to be text phrase from the author guidelines.

Deleted. 

Page 5, lines 205: the only published information on durability: there are many more references on meat (even without processed food). These should be added. E.g. Blome and Dietze 2011, McKercher et al., 1987

To our knowledge there are not ‘many’ more references on pork (perhaps on other meat) reporting primary research to determine the durability of ASFV in pork. We did not include reviews and risk analyses that refer to published work. McKercher et al (1987) is included and refers to Parma ham. We could perhaps have included McKercher et al (1978) but that refers to partly cooked canned hams and dried pepperoni and salami sausages, but no virus was obtained from the cooked product and neither the dried pepperoni nor salami sausages yielded virus after the required curing period, so the results do not add very much to the information provided.

An exhaustive search of the Web of Science, Scopus and Pubmed databases as well as Google scholar, failed to find any relevant publication (or any publication) by Blome & Dietze 2011, but if the reviewer could supply us with a copy we would be delighted to include it if it contains different information. 

Ok, I agree, Blome and Dietze is an unpublished FAO Project Report.

Blome S., Dietze K. Report on the stability of African swine fever virus strain “Armenia 2008” in different diagnostic materials after storage at different ambient temperatures. 2011. Unpublished FAO Project Report.

Page 5, line 210: reference 92 is a review, not the original literature on the R0

Certainly, but we considered the information to be relevant. Reviews can provide convenient summaries and analyses of primary information, as this one does; we use it as a reference to the fact that the spread of ASF can be either fast or slow, without having to cite a string of references on R0 as that is not the main topic of this review. 

ok

Page 5, line 215-217: not sure if this sentence belongs into the aforementioned context

Thank you for pointing this out. We have expanded it slightly and moved it to a separate paragraph, as it is not specific to a particular transmission route but is often the determinant of how the virus spreads regardless of the transmission route. 

Page 5, lines 219 ff: this is about large-scale commercial farms – which you did not want to write about (as mentioned further up).

It seemed appropriate to expand a little on why we did not include an in-depth discussion of biosecurity on large-scale commercial farms. We do not see any reason to remove this paragraph. 

I would still leave it out, as it is neither the focus of your review, and as you wrote you did not want to write about large-scale commercial farms. There are other concepts in your review which equivalently hold for large-scale commercial farms. So to me it seems inconsistent.

Page 5, line 225: “not specifically required” – required not at all, or specifically for ASF or for other diseases?

As the sentence states, not specifically required to prevent ASF.

Could you clarify this in the manuscript please?

Page 6, line 237: warthogs, domestic pigs and ticks

Ticks have been added, thank you.

Page 6, line 238: Pig-associated (which pigs? Domestic, warthogs?

Sentence altered to be clearer, thank you. 

Page 6, line 240: ticks .. are not a concern. (please give a reference)

The sentence has been changed to ‘not known to be a concern’. That is closer to the truth and does not need to be referenced – thank you for pointing it out. 

Page 6, line 251: long distance dispersal of flies? , but with much shorter distances for unfed and partially fed flies – than for fed flies?

Presumably, but the authors did not specify that so for us to do so would not be accurate citation. 

Ok – although an unclear writing in the original literature does not validate to do it in the review without making sure that it is not your unclear writing, but the problem of the original literature.

Page 6, table 2: please add references, as you did in table 1 and Page 7, table 3: please add references, as you did in table 1 – it is not clear if these are your opinions or if they are based on references.

References have not been added for the reasons provided in the general responses. We cannot attribute the recommendations to any particular reference, they are very widely reported and discussed. We did try adding references to table 3 but decided to remove them for that reason. 

As long as you make clear that the groupings are based on the authors experience then the issue of not including references is acceptable.

Page 7: from here onwards, the focus is very much on Africa, which is fine, but I would focus the whole manuscript on Africa, then it is more coherent and clear.

We have indicated why we prefer not to imply that this paper is only relevant for Africa, because it was largely inspired by the fact that whether outbreaks of ASF occur in Africa, Asia, the Pacific region or Eastern Europe, there is a predominance of backyard farms and village pigs among the epidemiological units affected (as reflected in the reports to OIE). We believe that the review is not relevant only to Africa, and that other regions can benefit from the long African experience of ASF. If we restrict it to Africa it may be written off as something unique to Africa and irrelevant anywhere else; if that was the case we would not have written it. 

I fully agree with the authors, that all countries can learn from the African experience. As long as you discuss the limitations of less data from other regions, it is acceptable.

Page 8, line 281: some cultures or communities: references missing

References added.

Page 8, lines 286: please add references

The example given is described in the reference cited and is unique to that reference, so we cannot add more references. We have checked the sentence again and it seems to be clear that we are citing a reference about an initiative in Timor-Leste. 

Ok

Page 8, lines 301/302: there are other references for feeding ASFV to pigs, not only the first description of Montgomery

Yes, but to provide a whole string of references for something as well-known as that seems unnecessary, and Montgomery was the first person (and one of the few) to undertake painstaking experiments with a range of foodstuffs fed in different forms to pigs and his findings have never been disproven, so we are happy to give him credit in the anniversary year of his providing the very first publication on ASF.

Ok, accepted

Page 8, line 307: not sure if the reference 118 is correct in this context.

It is correct and appropriate. 

Page 8, lines 324: please add references – this part of the manuscript it seems more like an opinion or a discussion, not like the parts before, where references are cited.

Line 324 includes the two relevant references, so presumably the reviewer is referring to the next paragraph (probably the additions made to the text so far have added a line). Much of what follows here is discursive, as there is not a lot of peer-reviewed literature on the subject, and most of it comes from the authors’ experience, which we think is legitimate to reflect in a review. Some of the information may come from mission reports by co-authors that are confidential so we cannot very well cite them although the information itself is not protected. 

Please make sure, that is the opinion of the authors and not published literature.

Page 9, lines 338ff: is this practical? Please give also a reference for this part.

It seems to be a fairly harmless recommendation and is practical in the sense that contamination of crops or vegetation with blood or faeces can be highly visible if the material is fresh enough to pose a threat. Reference 73 mentions this as a source of infection but makes no recommendations for avoiding the problem and the first author mentioned in a meeting that one backyard farmer had admitted to collecting grass from a wild boar deathbed site and feeding it to his pigs, although he saw that it was contaminated. We cannot share other peoples’ stories that they have not chosen to publish themselves but we believe we may take note of them and base upon them what we feel may be appropriate recommendations. 

Ok, accepted for macroscopical contamination, but in many cases the farmers do not see any contamination and to rinse with fresh water is not always feasible in poor resource settings.

Page 9, lines 340ff: scavenging pigs are outside, and most of the time you do not check the environment for carcasses to dispose of them.

That is so, and that is why we recommend as far as possible not to carelessly dispose of the carcasses of pigs. Obviously if wild boars are an issue there are several more reasons not to allow pigs to roam freely so this paragraph refers to domestic pigs in areas where the most likely carcass will be that of another pig. There are, incidentally, no documented cases of environmental contamination by the carcasses of African wild suids, which usually do not contain significant amounts of ASF virus. 

Could you please include this in the manuscript?

Page 9, line 348: please add reference

Another reference to unpublished data of one of the authors has been added.

Page 9, lines 351: what is wrong with not accepting remains as feed for pigs in situations, where people never felt hunger?

There is nothing wrong with that but if the pig died of ASF the uncooked remains should not be fed to pigs due to the probability of infection. The comment about people who have not felt hunger refers to the fact that even if they are horrified by people cooking and eating dead animals including those that have died of ASF it is not within their rights to disapprove of the practice. It is a bit like disapproving of the cannibalism admitted to after the Andean plane crash – none of us know what we might do to survive under similar circumstances. 

I agree that we do not know how we will react in those situations, and just as I do not disapprove of this attitude, I do not disapprove of the attitude of not accepting remains as feed for pigs in situations, where people never felt hunger.

Page 9, line 362/363: improving diagnostic capabilities of farmers. What is meant by this?

Enabling farmers to recognise the signs of ASF to support early detection and reporting of outbreaks. It often achieves this, but can also prompt the panic selling of pigs. The sentence containing the phrase has been split for the sake of clarity. 

Could you please include this (recognising the signs of ASF)  in the manuscript? Otherwise it could mean that you give the farmers penside-tests, which would not feasible/suitable in poor-resource settings.

Page 10, line 393: there are additional references detailing this

Agreed, but there are so many references already that we tried to be selective, because this is not a systematic literature review and its main focus is not on pigs surviving individual infection under experimental conditions but on pigs surviving infection with virulent viruses under natural conditions. 

ok

Page 10, lines 405 ff: the origin of the piglets is not clear. Is it important for the review to detail, that the piglets were purported as being wild, but they were not? This part could be shortened.

The section has been shortened accordingly to remove unnecessary information.

The genetic evidence showed very clearly that the piglets were pure Sus scrofa and therefore there was no evidence of genetic inputs from Potamochoerus larvatus, which differs genetically from domestic pigs and wild boars. Further investigation of the farms where the animals were bought revealed that they were Eurasian wild boar-domestic pig crosses, but the local farmers called the wild boars by the same common name as they would use for bushpigs (in fact a direct translation into their language of ‘bush pig’). To explain all of that would make the paragraph longer, but we think that what is written even in the new shorter version makes the situation clear – we do know what the origin of the pigs was and we did not purport that they were wild, only that they were purported to result from hybridization with wild pigs (bushpigs), which turned out not to be the case.

thank you 

Page 10, line 426: it would be good to state what Vasco et al have reported

As this is stated in the previous paragraph, i.e. that inbred pigs, i.e. pigs that were more genetically homogeneous showed more resistance, we did not think it necessary to repeat it.

ok

Discussion:

Page 11, line 471: current outbreaks where? In which region? Thus it would be good to focus the paper only on Africa.

The sentence has been modified accordingly.

Page 11, line 484: delete the space between ‘species’ and ‘are’

Space deleted, thank you. 

Page 13, line 578: simply changed or ceased – please give a reference for this

References added.

Page 13, line 573: if you mention Covid 19 (presumably in order to use the hype of the moment) then please also add a reference for it

The reference to the COVID-19 pandemic had nothing to do with the hype of the moment, it simply in some countries including the country of two of the authors has provided magnificent examples of how human behaviour – or misbehaviour – can feed a pandemic. However, combing the plethora of literature that has emerged to find something that may not be there, but is simply on our television screens every day, is beyond the scope of this review and we have deleted it. 

Thank you.

Page 13, line 585: “the focus populations of this review”. It is not clear what you mean with the reference 113? Why do you add reference 113 if you talk about the current review?

The citation has been moved to the previous sentence, where we agree with you that it is more applicable, thank you. 

Page 13, line 589: which may appear revolutionary: which options appear revolutionary?

Recommending that people who in any case are consuming their pigs that become victims of ASF should cook them well may be considered revolutionary and so may genetically modified pigs, but on mature reflection we believe it better not to emphasise that aspect too much, thank you for assisting the thought processes on that, and for a thorough and thoughtful review.

I would disagree with the authors, that this is revolutionary, it makes fully sense. But ok, I agree, it might appear revolutionary but only the ones outside of resource-poor settings.

Author Response

Comments and Suggestions for Authors

Note – reviewer comments that were addressed satisfactorily according to the review during the first round have been deleted. Responses to second round comments and queries are presented in italics and underlined.

Responses to general comments

Thank you very much for your review of ‘alternative’ approaches. The question is, alternative approaches to what? Biosecurity is practiced all over the world in relation to ASF and in all pig production systems. Thus, biosecurity is not an alternative approach.

While we agree that biosecurity is practiced throughout the world in the modern pig farming sector, in the sector we are considering, mostly no measures to manage ASF effectively are in place. The problem furthermore is that the level of biosecurity recommended for modern, technically advanced pig farms is neither feasible nor necessary for smallholder farmers to keep their pigs safe from ASF. While in some of the EU member states smallholder farmers who cannot comply with the exacting biosecurity standards implemented on industrialized pig farmers have been forced to stop pig farming, this could hardly be applied to the millions of poor smallholder pig farmers in developing countries worldwide who depend on their pigs for various reasons, including as a very important source of income. We are therefore proposing a specific basic level of biosecurity as an alternative to (1) no management, (2) comprehensive and efficient vaccination to protect the pigs despite a lack of biosecurity and (3) enforcing a ‘’one size fits all’’ approach to biosecurity in all types of pig farms. Please note also that we refer to ‘’alternative and complementary’’ measures as we would agree with anyone who says that vaccination alone will not protect fully against ASF. 

Thank you very much for your answer. It would be good to include this reasoning into the manuscript

Respond round 2: We have added some sentences to the introduction to convey this reasoning more clearly

Furthermore, I would suggest to focus the paper on Africa only, as most of the literature cited in the review is focused on Africa. It would be feasible to add the other countries/regions in the discussion. In its current form, it seems that the review is mainly based on information from Africa, where the information is good and plentiful. But the situation of the smallholders in other regions of the world is not as detailed described as the one from Africa. Often it is not clear on which region the paragraphs or sentences are focused on.

We believe that what we are proposing would be suitable in a wide range of smallholder pig farms wherever they are as the knowledge and experience of the co-authors does extend beyond Africa, and we do not want this paper to be written off as just another paper about African problems that are not relevant elsewhere. However, we have added a sentence to explain why the focus and examples given are largely from Africa because ASF has been endemic in Africa for very much longer than on the other continents now affected.

Thank you very much for your answer. I fully agree that any country can learn a lot from the African experience, and most experience is coming from Africa – please include your reasoning into the manuscript and detail the limitations of having less data on other regions.

Respond round 2:This has been addressed briefly in the last paragraph of the introduction.

Furthermore, the discussion seems to focus on participatory methods, but there is hardly anything mentioned about this in the main text of the review. Either delete large parts of the discussion or add further references on participation into the main text.

The main text of the review presents the available biosecurity measures required to prevent ASF in smallholder pig farms and constraints for their implementation. The use of participatory methods applies to the entire spectrum of measures and it would have been tedious to introduce the concept into each of the sections that deal with a specific aspect of biosecurity. For many small-scale pig farmers education and training have proven sufficient to result in improved biosecurity, but in some contexts where this is not the case, particularly where the constraints are socio-cultural and not just financial or due to lack of information. Some additions and changes to make this clearer have been added on lines 202-03, 240, 270-72..

Not sure your answer: “it would have been tedious to introduce the concept..” is convincing. I will leave this up to the editor.

Respond round 2:To the editor: We have checked for appropriate additional places to mention participatory approaches and have done so accordingly

Tables 2 and 3 detail biosecurity measures, but it is not clear on what basis these were grouped – there are no references given.

The grouping of the measures in Tables 2 and 3 is based on the knowledge and experience of the authors. All of them are covered in the text and it does not seem necessary to add the references to the tables as well. 

Please detail the fact, that the knowledge and experience of the authors is the basis for grouping the measures in the manuscript.

Respond round 2: The headings to the tables have been modified accordingly.

in the title you talk about resource poor settings, this is not coming out in the manuscript.

Just about all the settings we write about are resource poor and I think it does come out in the paper as it is mentioned in both the introduction and the section on free-range pig-farming, and the fact that it is in the title should in fact be sufficient to inform the reader as to our target context.

Yes, you as the authors know that all the settings are about resource poor situations, but the reader needs more clarification in this.

Respond round 2: We have clarified this further in the introduction.

Specific comments

Pages 8 and 9 are more a personal discussion than a review

Re pages 8 and 9, discussion of the topics is necessary as they are important in the context, and references have been used when available, but we did not see this as purely a review of literature and we have included our own experience and opinions. 

That is fine, if you state is as such, as own experience and opinion!

Respond round 2: This has been done where appropriate.

Page 4, lines 162: are backyard pig farms not also included in production systems with free-ranging pigs?

Sometimes they may be, but many or most are not because the pigs are permanently enclosed. 

It would be good to include this into the manuscript

Respond round 2: The paragraph on backyard farms does mention that in some cases pigs may be partly free-roaming, in which case the previous paragraph also applies, so we think the issue is adequately covered.

Page 5, lines 205: the only published information on durability: there are many more references on meat (even without processed food). These should be added. E.g. Blome and Dietze 2011, McKercher et al., 1987

To our knowledge there are not ‘many’ more references on pork (perhaps on other meat) reporting primary research to determine the durability of ASFV in pork. We did not include reviews and risk analyses that refer to published work. McKercher et al (1987) is included and refers to Parma ham. We could perhaps have included McKercher et al (1978) but that refers to partly cooked canned hams and dried pepperoni and salami sausages, but no virus was obtained from the cooked product and neither the dried pepperoni nor salami sausages yielded virus after the required curing period, so the results do not add very much to the information provided.

An exhaustive search of the Web of Science, Scopus and Pubmed databases as well as Google scholar, failed to find any relevant publication (or any publication) by Blome & Dietze 2011, but if the reviewer could supply us with a copy we would be delighted to include it if it contains different information. 

Ok, I agree, Blome and Dietze is an unpublished FAO Project Report.

Blome S., Dietze K. Report on the stability of African swine fever virus strain “Armenia 2008” in different diagnostic materials after storage at different ambient temperatures. 2011. Unpublished FAO Project Report.

Respond round 2: Thank you for this clarification.

Page 5, lines 219 ff: this is about large-scale commercial farms – which you did not want to write about (as mentioned further up).

It seemed appropriate to expand a little on why we did not include an in-depth discussion of biosecurity on large-scale commercial farms. We do not see any reason to remove this paragraph. 

I would still leave it out, as it is neither the focus of your review, and as you wrote you did not want to write about large-scale commercial farms. There are other concepts in your review which equivalently hold for large-scale commercial farms. So to me it seems inconsistent.

Respond round 2: We have shortened the paragraph considerably in terms of commercial farming but have retained part of it to make the point that an alternative approach to what is applied on commercial farms is needed in resource-poor settings.

Page 5, line 225: “not specifically required” – required not at all, or specifically for ASF or for other diseases?

As the sentence states, not specifically required to prevent ASF.

Could you clarify this in the manuscript please?

Respond round 2: With the alterations to the paragraph this sentence is no longer there and the concept is captured in the text added.

Page 6, line 251: long distance dispersal of flies? , but with much shorter distances for unfed and partially fed flies – than for fed flies?

Presumably, but the authors did not specify that so for us to do so would not be accurate citation. 

Ok – although an unclear writing in the original literature does not validate to do it in the review without making sure that it is not your unclear writing, but the problem of the original literature.

Respond round 2: We have returned to the original reference and placed ‘partially fed and unfed’ between inverted commas to indicate that it is a direct quotation from the reference cited.

Page 6, table 2: please add references, as you did in table 1 and Page 7, table 3: please add references, as you did in table 1 – it is not clear if these are your opinions or if they are based on references.

References have not been added for the reasons provided in the general responses. We cannot attribute the recommendations to any particular reference, they are very widely reported and discussed. We did try adding references to table 3 but decided to remove them for that reason. 

As long as you make clear that the groupings are based on the authors experience then the issue of not including references is acceptable.

Respond round 2: As indicated above, the headings have been modified accordingly

Page 7: from here onwards, the focus is very much on Africa, which is fine, but I would focus the whole manuscript on Africa, then it is more coherent and clear.

We have indicated why we prefer not to imply that this paper is only relevant for Africa, because it was largely inspired by the fact that whether outbreaks of ASF occur in Africa, Asia, the Pacific region or Eastern Europe, there is a predominance of backyard farms and village pigs among the epidemiological units affected (as reflected in the reports to OIE). We believe that the review is not relevant only to Africa, and that other regions can benefit from the long African experience of ASF. If we restrict it to Africa it may be written off as something unique to Africa and irrelevant anywhere else; if that was the case we would not have written it. 

I fully agree with the authors, that all countries can learn from the African experience. As long as you discuss the limitations of less data from other regions, it is acceptable.

Respond round 2: That is mentioned in the introduction, and we have cited relevant available literature from the Asia-Pacific region and eastern Europe.

Page 9, lines 338ff: is this practical? Please give also a reference for this part.

It seems to be a fairly harmless recommendation and is practical in the sense that contamination of crops or vegetation with blood or faeces can be highly visible if the material is fresh enough to pose a threat. Reference 73 mentions this as a source of infection but makes no recommendations for avoiding the problem and the first author mentioned in a meeting that one backyard farmer had admitted to collecting grass from a wild boar deathbed site and feeding it to his pigs, although he saw that it was contaminated. We cannot share other peoples’ stories that they have not chosen to publish themselves but we believe we may take note of them and base upon them what we feel may be appropriate recommendations. 

Ok, accepted for macroscopical contamination, but in many cases the farmers do not see any contamination and to rinse with fresh water is not always feasible in poor resource settings.

Respond round 2: We have tried to make it clearer that this applies to visibly contaminated material. Agreed that saliva or urine contamination might not be easily visible, but the risk of unseen contamination is lower than the risk of gross contamination. We have added a sentence indicating that collecting forage in the vicinity of a carcass is not recommended (at all).

Page 9, lines 340ff: scavenging pigs are outside, and most of the time you do not check the environment for carcasses to dispose of them.

That is so, and that is why we recommend as far as possible not to carelessly dispose of the carcasses of pigs. Obviously if wild boars are an issue there are several more reasons not to allow pigs to roam freely so this paragraph refers to domestic pigs in areas where the most likely carcass will be that of another pig. There are, incidentally, no documented cases of environmental contamination by the carcasses of African wild suids, which usually do not contain significant amounts of ASF virus. 

Could you please include this in the manuscript?

Respond round 2: A short paragraph referring to the situation in wild suids has been added.

Page 9, line 362/363: improving diagnostic capabilities of farmers. What is meant by this?

Enabling farmers to recognise the signs of ASF to support early detection and reporting of outbreaks. It often achieves this, but can also prompt the panic selling of pigs. The sentence containing the phrase has been split for the sake of clarity. 

Could you please include this (recognising the signs of ASF)  in the manuscript? Otherwise it could mean that you give the farmers penside-tests, which would not feasible/suitable in poor-resource settings.

Respond round 2: We have replaced ‘diagnostic capability’ with ‘ability to recognize the signs of ASF’.

Page 13, line 589: which may appear revolutionary: which options appear revolutionary?

Recommending that people who in any case are consuming their pigs that become victims of ASF should cook them well may be considered revolutionary and so may genetically modified pigs, but on mature reflection we believe it better not to emphasise that aspect too much, thank you for assisting the thought processes on that, and for a thorough and thoughtful review.

I would disagree with the authors, that this is revolutionary, it makes fully sense. But ok, I agree, it might appear revolutionary but only the ones outside of resource-poor settings.

Respond round 2: Thank you to the reviewer, we really appreciate a very thorough and supportive review that we believe has helped us to materially improve the manuscript.

Submission Date

31 December 2020

Date of this review

27 Jan 2021 16:56:53
